# An Environmentally Friendly Approach for the Release of Essential Fatty Acids from Cereal By-Products Using Cellulose-Degrading Enzymes

**DOI:** 10.3390/biology11050721

**Published:** 2022-05-08

**Authors:** Vitalijs Radenkovs, Karina Juhnevica-Radenkova, Jorens Kviesis, Anda Valdovska

**Affiliations:** 1Processing and Biochemistry Department, Institute of Horticulture, LV-3701 Dobele, Latvia; karina.juhnevica-radenkova@llu.lv; 2Research Laboratory of Biotechnology, Latvia University of Life Sciences and Technologies, LV-3002 Jelgava, Latvia; anda.valdovska@llu.lv; 3Laboratory of National Centre of Forest and Water Research, Faculty of Geography and Earth Sciences, University of Latvia, LV-1004 Riga, Latvia; jorens.kviesis@lu.lv; 4Faculty of Veterinary Medicine, Latvia University of Life Sciences and Technologies, LV-3004 Jelgava, Latvia

**Keywords:** biorefining, by-products, enzymatic hydrolysis, essential fatty acids, green-extraction, lipids, sustainability

## Abstract

**Simple Summary:**

Fatty acids, especially the representatives of essential mono- and polyunsaturated ones, play an important role in the human organism, as they are involved in the regulation of the immune and central nervous systems. Whole grain products are considered to be a rich source of multiple health-promoting phytochemicals, including fatty acids, where polyunsaturated fatty acids take prevalence over saturated ones. To improve the milling performance of grain and ensure the high-quality standards of flour, technologies presently utilized within the milling process, e.g., dehulling and debranning, generally aim at removing outer layers of cereal grain and result in substantial reduction of valuable nutrients along with loss of functionality. In spite of the relative abundance of valuable compounds in cereal bran, currently less than 10% of produced bran is used in the food industry. To valorize cereal bran for food and pharmaceutical applications, additional pre-treatment is required.

**Abstract:**

The main intention of the present work was to investigate the ability of cellulose-degrading enzymes (C-DE) to release fatty acids (FAs) from complex matrices of cereal by-products during enzymatic hydrolysis (EH). For this purpose, three types of cereal bran (CB), i.e., wheat, rye, and oat, were used as lignocellulose substrates for three commercially available hydrolytic enzymes, i.e., Viscozyme L, Viscoferm, and Celluclast 1.5 L. The yield and composition of FAs after EH were assessed and compared with those obtained after either conventional Soxhlet extraction or after alkaline-assisted hydrolysis (A-AH) with 10% KOH in 80% MeOH and subsequent liquid–liquid extraction. The experimental results demonstrated that up to 6.3% and 43.7% higher total FA yield can be achieved by EH of rye bran using Celluclast 1.5 L than by A-AH and Soxhlet extraction, respectively. However, the application of Viscoferm for EH of wheat bran ensured up to 7.7% and 13.4% higher total FA yield than A-AH and Soxhlet extraction, respectively. The concentration of essential linolenic acid (C18:3) in lipids extracted after EH of rye bran with Celluclast 1.5 L was up to 24.4% and 57.0% higher than in lipids recovered by A-AH and Soxhlet extraction, respectively. In turn, the highest content of linolenic acid in wheat bran lipids was observed after EH with Viscoferm and Viscozyme L, ensuring 17.0% and 13.6% higher yield than after A-AH, respectively. SEM analysis confirmed substantial degradation of the CB matrix promoted by the ability of C-DE to act specifically on glycosidic bonds in cellulose and on xylosidic bonds in arabinoxylans, arabinans, and other arabinose-containing hemicelluloses. Structural alterations in cell integrity greatly contributed to the release of bound FAs and their better transfer into the extraction solvent. It has been shown that the proposed process of EH can be used for the efficient release of FAs from the CB matrix more sustainably and with a safer profile, thereby conveying greener production of FAs for certain purposes.

## 1. Introduction

Technologies presently utilized for the pretreatment of raw materials, i.e., pasteurization, sterilization, cooling, freezing, refining, defatting, deproteinization, debranning, etc., generally aimed at ensuring the safety of the end products result in the substantial removal of valuable nutrients along with loss of functionality [1]. Cereal bran (CB) can be mentioned as one of the examples that, despite the relative abundance of dietary fiber, are sorted out along the grain milling process and later disposed of as waste or utilized rather inefficiently [2]. Underestimation of CB as a food ingredient is conditioned by several drawbacks that are manifested in the form of negative influence on technological processes and impairments of sensory quality of the end product by providing bitterness [3], reducing loaf volume, and contributing to textural changes, e.g., porosity and elasticity [4]. The reports of Jefremova et al. [5] and Aravind et al. [6] demonstrated that along with the increase of dietary fiber in the prepared products, the inclusion of wheat bran in the formulation of more than 7% and 30%, resulted in physico-chemical and sensory quality losses of bread and pasta, respectively. The adverse effects of CB were highlighted by Lebesi and Constantina [7], noting an increase in crumb firmness and the darkening of cupcakes made with the addition of 10–30%, leading also to lower sensory scores compared to control samples. These observations have further been reinforced by Grigor et al. [8], conducting a meta-analysis of sensory acceptance of fiber-enriched cereal foods. It was revealed that adding as little as 2 g of fiber to the formulation resulted in a moderate–large reduction of overall acceptability, flavor, and appearance in most elaborated products.

Due to palatability, refined food products rather than whole-grain products rich in dietary fiber are more appreciated by consumers [9]. There is a body of clinically proven health benefits, however, supporting that whole-grain diets promote reductions in body weight, fat loss, systolic blood pressure, and LDL cholesterol levels [10,11,12,13,14]. Given the above, new strategies aimed to process CB are needed to be elaborated that would attract consumers with innovative food products enriched with nutrients that were lost during processing while not affecting the sensory qualities of it [15].

There have been various pretreatment strategies developed to date aimed at the valorization of CB to food and pharmaceutical applications, the vast majority of which, however, imply chemical modifications of plant cell walls by using solvents and catalysts that are potentially harmful to operators and the environment [16,17]. Furthermore, physical and physico-chemical pretreatments that have been demonstrated, such as mechanical extrusion [18] and steam explosion [19], are materially costly technologies requiring high initial investments [20]. Contrary to these methods, biorefining of lignocellulosic waste materials due to the simplicity in operational and process conditions has been receiving tremendous interest among researchers as a potential strategy suitable for the enhancement of physical and physico-chemical features as well as the production of biomolecules and chemicals [21,22]. Biorefining by EH is an environmentally friendly alternative that may ensure sustainability and circularity through the utilization of renewable raw materials and food-grade enzymes for obtaining high-added-value products and energy from biomass [23].

Owing to the relative abundance of phenolic compounds, which in CB are present mainly in non-extractable forms [24,25], the effectiveness of EH to release ferulic acid has repeatedly been proven [26,27]. Furthermore, the availability of dietary fiber, i.e., cellulose and hemicellulose [28,29], makes this material suitable for the production of xylooligosaccharides [30] and fermentable sugars, compounds that are in demand for making functional food products and bioethanol, respectively [31,32]. The ability of a commercial enzyme mixture and *Escherichia coli* JM109 to produce biovanillin from wheat-bran-derived ferulic acid was demonstrated by Gioia et al. [33]. Furthermore, the single-step optimized process of EH utilizing the filamentous fungi *Rhizopus oryzae* wild1.22 ensured the yield of fumaric acid up to 20.2 g L^−1^ from acid-pretreated wheat bran hydrolysates [34].

The EH as a type of CB pretreatment has been well-studied, demonstrating the capability of hydrolytic enzymes to modify plant cell walls with the simultaneous release of valuable hydrophilic compounds. Less explored, however, is the influence of hydrolytic enzymes on the yield and chemical profile of the lipophilic fraction (FAs) obtained upon direct hydrolysis of CB, in particular wheat, rye, and oat. EH as a novel plant pretreatment approach for the extraction/release of FAs has been shown in a few reports by [35,36,37], where the effectiveness of hydrolytic enzymes in altering the cell-wall structure and the release of FAs was confirmed by observing a higher yield and qualitative profile of oils than from untreated matrices. Given that the majority of FAs (mainly in the form of triacylglycerols) in grain kernels are located in aleurone and starchy endosperm fractions [38,39,40], it is hypothesized that the designed process of EH can be used for the efficient release of FAs from the bran matrix in a more sustainable way and with a safer profile, and thereby it may represent the further green production of FAs for certain purposes.

The main intention of the present study was to study the ability of C-DE to release FAs from complex matrices of cereal by-products during EH.

## 2. Materials and Methods

### 2.1. Plant Material

Three types of commercial food-grade CB samples were obtained from a local supplier, “Voldemars Ltd.”, separated as wheat (*Triticum aestívum* L.), rye (*Secale cereále* L.), and oat (*Avena sativa* L.). According to morphological assessment, hydrological layers such as the inner pericarp (tube cells, cross cells), outer pericarp, aleurone, and hyaline layers with attached starch granules and seed coat (testa) were distinguished in the bran samples. The approximate compositions of the CB samples are shown in Table 1.

### 2.2. Plant Material Preparation for Alkaline and Enzymatic Hydrolysis and Analysis of Hydroxycinnamates

Each CB sample before EH was ground to reach Ø 0.5 mm particle size using the water-cooled “KN 295 Knifetec™” rotor mill (FOSS, Hilleroed, Denmark). Inactivation of native microorganisms and enzymes was done by mixing CB samples with double distilled water (DDW) at a ratio of 1:10 *w/v* in 50 mL reagent bottles with screw caps (VWR™ International, GmbH, Darmstadt, Germany), followed by subjecting them to autoclaving using a “Raypa, AES 110” (Barcelona, Spain) digital autoclave with counter-pressure for 10 min at a temperature of 121 ± 1 °C and a counter pressure of 2.0 Pa. After thermal conditioning, the liquid fraction was decanted, while solids were freeze-dried using a “Christ Alpha 1-2 LDplus” freeze-drying system (Osterode, Germany) at −51 ± 1 °C under a vacuum of 0.070–0.080 mBar for 72 h. Dried solids were packed in polypropylene zip-lock silver bags (high-density polyethylene polymer, density 3 mm, Impak Co., Los Angeles, CA, USA) (200 g in each) and stored at a temperature of −18 ± 1 °C until further analysis and used for a maximum of two weeks. Moisture content was analyzed gravimetrically, as proposed by Ruiz [41].

### 2.3. Chemicals and Reagents

A standard solution containing a mixture of C_4_-C_24_ fatty acid methyl esters (FAMEs) with a purity of ≥99.0% was acquired from Sigma-Aldrich Chemie Ltd. (St. Louis, MO, USA). Sodium hydroxide (NaOH), potassium hydroxide (KOH), citric acid (C_6_H_8_O_7_), sodium citrate dihydrate (C_6_H_5_Na_3_O_7_·2H_2_O), phenolphthalein (C_20_H_14_O_4_), and 0.5 M trimethylphenylammonium hydroxide solution (CH_3_)_3_N(OH)C_6_H_5_ (TMPAH) in methanol (MeOH) for GC derivatization were of reagent grade and were obtained from Sigma-Aldrich Chemie Ltd. HPLC grade MeOH was purchased from Sigma-Aldrich Chemie Ltd. Petroleum ether (puriss p.a., ≥99.9%, boiling point 40–60 °C) and diethyl ether ((C_2_H_5_)_2_O) (puriss p.a., ≥99.5%) were obtained from Chempur (Piekary Śląskie, Silesia, Poland). The ultrapure water was produced using the reverse osmosis PureLab Flex Elga water purification system (Veolia Water Technologies, Paris, France).

### 2.4. Enzymes

Commercially available food-grade C-DE preparations was provided in kind by the company Novozymes^®^ (Bagsvaerd, Denmark) for laboratory purposes. Each enzyme preparation selected was used in this study individually rather than as a mixture. A list of enzymes with their main activities is depicted in Table 2.

### 2.5. Soxhlet Extraction

Soxhlet extraction was done using the method of Abdolshahi et al. [42] with slight modification. Triplicate samples of about 10 g of freeze-dried and finely ground CB were accurately weighted in extraction cellulose cotton thimbles (Whatman single thickness, 25 × 100 mm) (VWR™ International, GmbH, Darmstadt, Germany). Further, the thimbles were placed inside the extraction chambers and subjected to Soxhlet extraction using the B-816 (BÜCHI Labortechnik AG, Flawil, Switzerland) system, which is fully automated. The extraction of the lipophilic fraction was done for 6 h using petroleum ether as the extraction solvent. Sufficient heat (heating plate temperature 195 °C) was used to give about 10 cycles of solvent per h. In the final preparatory stage, collected lipophilic fractions were dried for 30 min to release the solvents from the extraction beakers. To determine the yield of crude lipids, the collected samples were placed in a desiccator to cool and then weighted. Then, the collected dry lipids were re-dissolved in 3 mL of petroleum ether and filtered through a polytetrafluoroethylene hydrophobic (PTFE) membrane filter with a pore size of 0.45 µm (VWR™ International, GmbH, Darmstadt, Germany). The obtained filtrates were quantitatively transferred to 20 mL scintillation glass vials (Kimble^®^ DWK Life Sciences, Millville, NJ, USA) and thereafter flushed with N_2_ for 10 min to complete dryness. Dry residues were stored at a temperature of −18 ± 1 °C, until further analysis and use, a maximum of five weeks.

### 2.6. Preparation of the Lipid Fraction by Enzyme-Assisted Hydrolysis of Wheat, Rye, and Oat Bran with Subsequent Liquid–Liquid Extraction

The EH of CB samples utilizing biocatalysts was performed in an “SW23” water bath, with a capacity of 20.0 L, and a horizontal shaking and thermostatic and temporal control system (Julabo^®^, Saalbach-Hinterglemm, Germany). The conditions for each enzyme were chosen individually based on Novozymes^®^ guidance and following the protocol described by Juhnevica-Radenkova et al. [26]. The EH of non-starch polysaccharides (N-SPs), i.e., cellulose and hemicellulose, was accomplished using three commercially available enzyme complexes, i.e., Viscozyme L, Viscoferm, or Celluclast 1.5 L. For this purpose, 30 mL 0.5 M sodium citrate buffer with a pH of 4.6 containing 6 FBG mL^−1^ of endo-1,3-(1,4)-β-D-glucanase (Viscozyme L or Viscoferm), or 10 EGU mL^−1^ of endo-1,4-β-D-glucanase (Celluclast 1.5 L) was added to 3 g of each CB sample. The mixture was then vortexed for 2 min using the “ZX3” vortex mixer (Velp^®^ Scientifica, Usmate Velate, Italy) and incubated for 48 h in a water bath at 44 ± 1 °C and 100 rpm. After EH, to terminate the reaction, the obtained hydrolysates were subjected to thermal processing in a water bath for 10 min at 99 ± 1 °C. The extraction of the lipophilic fraction was accomplished by liquid–liquid phase separation using petroleum ether as a solvent. After being cooled down to ambient temperature (22 ± 1 °C), hydrolysates were quantitatively transferred to Falcon 50 mL conical centrifuge tubes (Sarstedt AG & Co. KG, Nümbrecht, Germany). Then, 10 mL of petroleum ether was added to each tube, followed by vortex-mixing for 1 min. Separation of the layers was done by centrifugation at 4500 rpm (3169 × *g*) for 10 min in a “Sigma, 2-16KC” centrifuge (Osterode, Germany). The top petroleum ether layer was separated and collected. The extraction procedure was repeated three times. The resulting lipophilic fraction (30 mL) was further evaporated using a “Laborota 4002” rotary evaporator (Heidolph, Swabia, Germany) at 65 °C, and the dry fraction was then re-dissolved in 3 mL of petroleum ether and filtered through a PTFE membrane filter with a pore size of 0.45 µm. The filtrates were quantitatively transferred to 20 mL scintillation glass vials and then subjected to drying under a gentle stream of N_2_ to complete dryness. Dry residues were kept for a maximum of five weeks at a temperature of −18 ± 1 °C until further analysis and use.

### 2.7. Preparation of the Lipid Fraction by Alkaline-Assisted Hydrolysis of Wheat, Rye, and Oat Bran with Subsequent Liquid–Liquid Extraction

For the destruction purposes of the CB matrix and the release of bound forms of FAs, 10% (*w/v*) KOH dissolved in 80% MeOH (*v/v*) was used. In the supremacy of MeOH, this approach allows the process of hydrolysis and the release of FAs to be conducted more efficiently. Triplicate samples of 3.0 ± 0.1 g of ground CB were weighed in 50 mL reagent bottles with screw caps. For the hydrolysis, 30 mL of prepared methanolic KOH was added to each CB sample, and the mixture was subjected to incubation in a “TW8” (Julabo^®^, Saalbach-Hinterglemm, Germany) water bath at 65 °C for 3 h. After hydrolysis, the release of FAs from the salt form was done by shifting the pH of the medium from alkaline to acidic by adding 6 M HCl until the pH was 2.0 (10 mL). The extraction of the lipophilic fraction was done using liquid–liquid phase separation as described above (please refer to Section 2.6).

### 2.8. Preparation of Fatty Acids for GC/MS Analysis Using 0.5 M TMPAH Solution

The 0.5 M TMPAH solution in MeOH reagent was utilized as a methylation agent. The methylation of polyfunctional groups to obtain volatile FAME derivatives was performed according to the protocol provided by Radenkovs et al. [21] with modifications. Briefly, 0.5 µL of the lipid fraction was mixed with 10 µL of 1% C_20_H_14_O_4_:EtOH (1:99 *w/v*), 70 µL of 0.5 M TMPAH reagent, and 919.5 µL of MeOH:(C_2_H_5_)_2_O (50:50 *v/v*) in a 2 mL chromatographic vial. The resulting permanent pink mixture was vortex-mixed for 1 min and kept in a GC oven at 60 °C for 30 min. Further, eliminating the mixture cooling step, the sample was injected into the heated injection port of the chromatograph, where the TMPAH salts were subjected to pyrolysis with subsequent transformations to their respective methyl esters.

### 2.9. The GC Conditions for FAME Analysis

The analysis of FAMEs was carried out on a PerkinElmer, Inc. “Clarus 600” system (Waltham, MA, USA) equipped with a “Clarus 600 C” quadrupole analyzer mass-selective detector (Waltham, MA, USA). The chromatographic separation of FAMEs was done using a “Trace™ TR-FAME” (Thermo Fisher Scientific, Waltham, MA USA) column with a cyanopropylphenyl-based stationary phase (50 m × 0.22 mm, sorbent thickness—0.25 μm) specifically designed for the separation of *cis*- and *trans*-isomers of FAMEs (Figure 1). The injector temperature was set to 280 °C; automatic injection using an autosampler, injection volume 0.5 μL; split ratio 4:1. The initial oven temperature was maintained at 70 °C for 2 min, then raised to 150 °C (rate of 20 °C min^−1^), then increased to 250 °C (rate of 4 °C min^−1^). Helium (ultra-high purity 5.0 grade—99.9%) was used as a carrier gas at the constant flow rate of 1.0 mL min^−1^. The total separation time was 31.00 min. The analysis was performed in triplicate.

### 2.10. The MS Conditions for FAME Detection

Detector mode: Electron impact ionization was at 70 eV; ion source temperature: 230 °C; inlet temperature was 250 °C; capture time starting from 6.5 min (1.7 scan s^−1^); ion multiplier: 240 V; and ion m/z interval: 41–500 atom mass units (AMU) for FAMEs.

### 2.11. The HPLC-RID Conditions for Carbohydrates Analysis

Quantitative analysis of mono- and disaccharides in hydrolysates after EH was accomplished on a Waters Alliance HPLC system (model No. e2695) coupled to a 2414 RI detector and a 2998 column heater (Waters Corporation, Milford, MA, USA) following the methodology described by Juhnevica-Radenkova et al. [26].

### 2.12. Scanning Electron Microscopy (SEM)

The morphology of untreated control and EH CB was analyzed by SEM using a Tescan Mira/LMU scanning electron microscope (Brno-Kohoutovice, Czech Republic) according to the method proposed by Juhnevica-Radenkova et al. [26].

### 2.13. Statistical Analysis

The results obtained are shown as means ± standard deviation of the mean from three replicates (*n* = 3). A *p*-value of ≤ 0.05 was used to denote significant differences between mean values determined using one-way analysis of variance (ANOVA) and Duncan’s multiple range test performed using IBM^®^ SPSS^®^ Statistics version 20.0 (SPSS Inc., Chicago, IL, USA).

## 3. Results and Discussion

### 3.1. Structural Changes in Wheat, Rye, and Oat Bran Morphology Induced by Cellulose-Degrading Enzymes

The assessment of morphology and microstructure of native CB samples revealed uniform homogeneity of the epidermal layer with no obvious signs of cracking or fractures (Figure 2A). Histological layers such as the outer and inner pericarp (outer fraction), and the cross and tube cells (intermediate fraction) with tight adherence to the walls of the aleurone layer (inner fraction) were distinguished by SEM analysis (Figure 2B). Two fractions of starch granules were observed in the starchy endosperm of CB samples (Figure 2C). Starch granules spherical in shape with a size smaller than 10 μm were used, taking prevalence over disk-shaped granules with sizes of 15–35 μm, making a strong carcass and holding the integrity of the CB matrix.

However, further analysis revealed clear and extensive decomposition of CB N-SPs caused by EH of CB for 48 h with the multi-enzyme complex Viscozyme L. The EH resulted in the partial fracture of the epidermal layer of CB, and the opening cross and tube cell (helix) microfibers can be seen in Figure 2E. Performing EH, visible void spaces in the testa (seed coat) and nucellar tissues with approximate holes sizes of 20–30 μm were revealed, which are depicted in Figure 2F. Similar epidermal layer degradation and disruption of the cell integrity was observed in rye bran samples subjected to 48 h EH with Viscozyme L, indicating the equal hydrolytic performance of the enzymes selected (Figure 2H). A similar observation was made by Zhang et al. [22], noting the ability of endo-1,4-β-xylanase and endo-1,4-β-D-glucanase (cellulase) enzymes to modify wheat bran structure during complex EH and as a consequence better physicochemical and functional properties. The observed alterations in the CB microstructure are associated foremost with the ability of cellulolytic and xylanolytic enzymes to act specifically on endo-1,3-(1,4)-β-D-glycosidic bonds in cellulose and on 1,4-β-D-xylosidic bonds in arabinoxylans, arabinans, and other arabinose-containing hemicelluloses, releasing relatively shorter polysaccharides or oligosaccharides composed of glucose, xylose, or arabinose [26].

Given that the majority of FAs are distributed across the aleurone layer and starchy endosperm fraction, the action of C-DE would affect the release of subcellular organelles called oil bodies and thereby promote the mass transfer of FAs into the extraction solvent. This statement could be reinforced by an observation made by Kaseke et al. [34], revealing the ability of a hydrolytic enzyme cocktail, composed of Pectinex Ultra SPL, Flavourzyme 100 L, and cellulase in equal proportions to enhance the yield of oil, carotenoids, and phenolic compounds from EH pomegranate seeds.

### 3.2. Release of Mono- and Disaccharides from Wheat, Rye, and Oat Bran Using Enzyme-Assisted Hydrolysis

Based on the datasheets ensured by Novozymes^®^ guidance, the enzyme preparations Viscozyme L and Viscoferm, selected for EH of CB, are multi-enzyme complexes consisting of such hydrolases as cellulase, xylanase, and arabinofuranosidase that under favorable conditions promote the hydrolysis of cellulose, hemicellulose, and β-glucans present in CB. The simultaneous release of glucose, arabinose, xylose, maltose, and galactose can be achieved due to the breakdown of glycosidic bonds in these N-SPs [43,44]. Therefore, the efficiency of EH in this study was assessed by determining the content of sugar monomers and dimers individually in CB that underwent hydrolysis with three C-DE for 48 h (Table 3). The results showed that glucose, xylose, arabinose, and fructose are the main end-products released after EH of wheat, rye, and oat bran samples. The number of sugars released was found to be the bran type and hydrolytic enzyme-dependent. The prevalence of glucose over other sugars was observed in all CB hydrolysates. The glucose concentration fluctuated in the range of 7.9–47.6 g per 1000 mL^−1^ of bran hydrolysates, with oat bran having the highest content and wheat bran the lowest. A considerably higher amount of glucose in oat bran hydrolysates is due to the presence of β-glucans, which alongside cellulose ensures the release of glucose monomers by endo-1,3-(1,4)-β-D-glucanase. Viscozyme L displayed superior hydrolytic performance since the yields of glucose monomers from wheat, rye, and oat bran were 58.4–126.6%, 163.4–58.4%, and 158.7–195.6% higher than those released after EH with Celluclast 1.5 L and Viscoferm, respectively. Higher glucose yield was conditioned by the composition and activity of the hydrolytic enzymes presented in the Viscozyme L preparation, which altogether promoted solubilization of cellulose and β-glucans and the rise of sugar monomers and dimers. This observation has been already confirmed in previous studies by Radenkovs et al. [21] and Bautista-Expósito et al. [45] working on CB hydrolysis with hydrolytic enzymes. In addition, the advantage of Viscozyme L over Celluclast 1.5 L was highlighted by Gama et al. [46], pointing to an increase in glucose level in the hydrolysates of apple pomace that underwent EH.

The presence of xylose and arabinose monomers (except for arabinose in oat after EH with Celluclast 1.5 L) was confirmed in all bran hydrolysates, the range of which fluctuated, respectively, from 0.8 to 8.7 g 1000 mL^−1^ and from 0.8 to 6.2 g 1000 mL^−1^. However, the ambiguity of the results obtained should be noted, since the highest yield of xylose in all bran hydrolysates was observed after EH with Celluclast 1.5 L, the enzyme which based on our knowledge has no xylanolytic activity. The highest yield of arabinose was detected in rye bran hydrolysates after EH with Viscozyme L, while no significant differences (*p* ≤ 0.05) were found between arabinose values in wheat and oat bran hydrolysates and enzymes used.

Assessment of the total content of sugars released demonstrated that 1000 mL^−1^ wheat, rye, and oat bran hydrolysates contained up to 28.6 g, 48.6 g, and 52.9 g of sugars, respectively. In general, the highest content of total sugar was observed after performing EH with Viscozyme L, except for wheat bran, where no significant (*p* ≤ 0.05) difference between Viscozyme L and Celluclast 1.5 L was revealed. Oat bran hydrolysates represented the highest total sugar content that was up to 84.9% and 8.8% higher in comparison to wheat and rye bran hydrolysates, respectively. A relatively high content of sugars is associated with a higher amount of starch in oat bran rather than with cellulolytic and xylanolytic activities of enzymes. Thermal processing that was applied to terminate the catalytic reaction of enzymes after EH presumably contributed to the partial degradation of glycosidic 1,4-α bonds in the starch polymer to its glucose monomer [47].

The concept of the Green Deal proposed by the European Commission could be supported by taking advantage of the ability of hydrolytic enzymes to release fermentable sugars more sustainably. Their further exploitation can be done by the production of bioethanol, which would foster the use of grain by-products more efficiently and contribute to a circular economy [19,48].

### 3.3. Effect of Alkaline- and Enzyme-Assisted Hydrolysis on Recovery of Lipids from Wheat, Rye, and Oat Bran 

Total lipids in the tested CB obtained by using liquid–liquid extraction following A-AH varied in the range of 4.0–7.5%, with rye bran having the lowest amount and oat bran the highest (Figure 3). The amount of lipids recovered from CB was in line with those reported by Kamal-Eldin et al. [49]. The amount of lipids yielded after A-AH was 83.9%, 122.2%, and 17.2% higher than that extracted by the Soxhlet apparatus. In turn, the amount of lipids extracted from CB after EH fluctuated in the range of 1.4–5.6%. The EH of CB with Viscozyme L ensured up to 1.4–4.9% of lipids, which was 12.9–23.4% lower than that recovered by Soxhlet extraction. Similar to Viscozyme L, the yield of lipids after EH of CB by Celluclast 1.5 L reached 1.7–5.6%, which was 5.5–12.5% lower than that recovered by Soxhlet extraction.

However, the advantage of Viscoferm over other enzymes used in this study was confirmed by collecting an equal amount of lipids from rye bran and even a 9.7% higher amount from wheat bran compared with Soxhlet extraction. A higher lipid yield compared to other enzymes was obtained after EH of oats with Celluclast 1.5 L, though the extracted amount was 12.5% lower than that from Soxhlet extraction. As shown, the enzymes used displayed relative fluctuations in the hydrolytic performance of CB; however, clear superiority of the proposed EH in the case of wheat and rye CB over the Soxhlet apparatus was seen.

### 3.4. Effect of Alkaline- and Enzyme-Assisted Hydrolysis on Recovery of Fatty Acids from Wheat, Rye, and Oat Bran

The composition of FAs in lipids recovered from wheat, rye, and oat bran is depicted in Table 4, Table 5 and Table 6. In total, 18 FAs were identified and quantified, among which the dominance of linoleic acid (C18:2) from 37.6 to 60.9%, followed by palmitic acid (C16:0) from 17.6 to 20.3%, oleic acid (C18:1) from 11.7 to 40.2%, linolenic acid (C18:3) from 0.8 to 4.9%, and stearic acid (C18:0) from 1.1 to 1.9% was found. The results are consistent with those of Narducci et al. [50], indicating a fairly similar descending order of FA content recovered from durum wheat grains.

Among the saturated FAs (SFA), the prevalence of palmitic acid was observed in all bran samples and pre-treatment types applied. However, the highest concentration was revealed in oat, followed by wheat and rye bran lipids, corresponding to 137.0–150.5 mg g^−1^, 127.0–143.9 mg g^−1^, and 90.2–125.5 mg g^−1^, respectively. Up to 13.3% and 12.5% better palmitic acid release from wheat bran was achieved by subjecting the bran samples to EH with Viscozyme L and Viscoferm compared to the Soxhlet type extraction, respectively. While a considerably higher yield of palmitic acid was recovered from rye bran samples after A-AH and EH with Celluclast 1.5 L, the values corresponded to an increase of 39.1% and 34.7% compared with Soxhlet extraction, respectively. However, neither EH nor A-AH had a tangible effect on the release of palmitic acid from the oat bran sample.

The concentration of oleic acid, the representative of monounsaturated FAs (MUFA) in wheat, rye, and oat bran lipids, varied in the range of 104.5–121.3 mg g^−1^, 64.2–87.2 mg g^−1^, and 266.6–343.9 mg g^−1^, respectively, though the results were inconsistent considering the yield of oleic acid as a function of the enzyme and type of bran. As seen, the multi-enzyme complex Viscozyme L in the case of wheat bran samples demonstrated better hydrolytic performance, since up to 16.1% and 14.3% higher yields of oleic acid were obtained compared with the Soxhlet extraction and A-AH hydrolysis, respectively. However, considering the amount of oleic acid in wheat lipids using other enzymes, no statistically significant difference (*p* ≥ 0.05) in the content of this FA was observed. The superior efficiency of hydrolysis and release of oleic acid was noticed after EH of rye bran with Celluclast 1.5 L. As seen, up to 35.7% and 14.0% higher yields were reached compared with Soxhlet extraction and A-AH hydrolysis, respectively. The amount of oleic acid released by the activity of Celluclast 1.5 L was found to be significantly (*p* ≤ 0.05) higher than that obtained after A-AH or EH with Viscozyme L and Viscoferm. It is worth noting, though, that the release of oleic acid was not correlated with the amount of sugars in hydrolysates, presumably revealing that this FA in the rye bran matrix presented as a glycolipid complex where FA attached to carbohydrates by a 1,4-β-D-glycosidic bond. The release of oleic acid from the rye bran matrix is most likely associated with glycoside hydrolase activity (endo-1,4-β-D-glucanase), which was more intense in Celluclast 1.5 L compared to Viscozyme L and Viscoferm. A similar observation has been made by Radenkovs et al. [21], highlighting the superiority of Celluclast 1.5 L in a more substantial performance of releasing polyunsaturated FAs over other hydrolytic enzymes. The obtained results indicate that a rather high side activity of triacylglycerol ester hydrolases likely is present in Celluclast 1.5 L along with cellulolytic activity that resulted in higher recovery of oleic acid through synergistic action. It is worth noting that the content of oleic acid in lipids recovered from oat bran samples using either A-AH or EH was found to be 3.3–22.5% lower compared with lipids recovered with the Soxhlet type extraction.

Among PUFA, the linoleic acid in CB lipids was found to be the dominant FA, making the largest contribution to the total content of PUFA in CB lipids. The concentration of linoleic acid varied in the range of 272.0–421.4 mg g^−1^, with wheat and rye bran having the highest concentration and oat bran having the lowest. The obtained results are in line with data from [51], highlighting the supremacy of wheat lipids over barley, rice, sorghum, and oats. The highest recovery of linoleic acid from CBs was achieved using EH with Celluclast 1.5 L, followed by Viscoferm, and Viscozyme L. Up to a 49.1% and 10.7% higher release of linoleic acid was achieved after EH of rye bran with Celluclast 1.5 L compared with Soxhlet extraction and A-AH, respectively. However, better recovery of linoleic acid from wheat bran was achieved utilizing either Viscoferm or Viscozyme L, where up to a 14.3 and 11.7% increase in the amount was observed compared with Soxhlet extraction, respectively.

The content of linolenic acid in lipids extracted from CB samples was found in the range of 6.1–33.9 mg g^−1^, with rye bran lipids having the highest content and oat bran the lowest. Similar to oleic acid, the content of linolenic acid in rye bran lipids obtained after EH with Celluclast 1.5 L up to 57.0% and 24.4% higher than in lipids was recovered with the Soxhlet apparatus and after A-AH, respectively. The abundance of linolenic acid in wheat bran lipids was confirmed by obtaining up to a 17.0% and 13.6% higher content after EH with Viscoferm and Viscozyme L than with A-AH, respectively. The effectiveness of Viscozyme L over conventional solvent extraction has been highlighted by Díaz-Suárez et al. [52], pointing out on the safer profile of recovered oil from castor seeds (*Ricinus communis*) with an equal profile of FAs.

The total content of FAs in bran-derived lipids is hydrolysis-method-dependent and varied in the range of 481.3–860.6 mg g^−1^ (Table 4, Table 5 and Table 6). Without reference to the CB pretreatment methods, the superiority of oat bran in the content of total FAs, including MUFA and PUFA over wheat and rye bran, was revealed. The results are consistent with data obtained in an earlier study [53]. A relatively higher amount of FAs in oat bran than in other CB is explained by the presence of a starchy endosperm, which is known to be a source of lipids [54]. The advantage of Celluclast 1.5 L over other enzymes in the release of FAs from rye bran was established in this study. The experimental results demonstrated that up to a 6.3% and 43.7% higher yield of total FAs can be achieved using EH with Celluclast 1.5 L than after A-AH and with Soxhlet extraction, respectively. However, in the case of wheat bran, the highest FA yield was obtained after EH with Viscoferm. As shown, up to a 7.7% and 13.4% higher total FA yield than after A-AH and with Soxhlet extraction was ensured, respectively. In general, the higher glycoside hydrolase activity (endo-1,4-β-D-glucanase) declared for Celluclast 1.5 L led to a substantial and more selective breakdown of 1,4-β glycosidic bonds between N-SPs and lipids, which presumably promoted the release of FAs. Moreover, superior xylanolytic than cellulolytic activity of Viscozyme L and Viscoferm was also confirmed by Wikiera et al. [55], thereby explaining the relatively fewer contributions of these enzymes to the release of glycolipids.

The yield of FAs from the oat bran matrix using EH was observed to be the lowest, since up to a 3.8–17.6% lower amount of FAs was obtained compared to Soxhlet. However, the EH of oat bran with Celluclast 1.5 L made the most tangible contribution to the release of FAs, while Viscoferm had the lowest. Moreover, no substantial release of FAs was achieved even after the application of A-AH. It is assumed that a significantly lower content of FAs in oat bran lipids can be associated with its autoxidation reaction in the presence of oxygen and lack of natural antioxidants such as hydroxycinnamic and hydroxybenzoic acid derivatives abundantly present in wheat and rye bran hydrolysates [21].

## 4. Conclusions

The present study was undertaken to clarify the ability of C-DE to release FAs from complex matrices of CBs during EH. The results revealed that among the hydrolytic enzymes selected, the superiority of Viscozyme L can be highlighted, since the highest total sugar content in the rye and oat bran hydrolysates was observed after performing EH for 48 h, though no significant (*p* ≤ 0.05) difference in the content of total sugar was observed between Viscozyme L and Celluclast 1.5 L in wheat bran hydrolysates. Oat bran hydrolysates contained the highest amount of total sugar, and this amount was 87.6% and 8.9% higher than in wheat and rye bran hydrolysates, respectively. Structural alterations in the cell wall integrity were observed by SEM analysis, confirming that obvious signs of epidermal cracking or fractures were most evident after EH with Viscozyme L. The results showed that the action of C-DE promoted the release of subcellular organelles and the breakdown of glycosidic bonds between N-SPs and lipids in glycolipids, thus contributing to the better transfer of FAs into the extraction solvent. Among the hydrolytic enzymes tested, the advantages of Celluclast 1.5 L and Viscoferm were highlighted. Up to a 6.3% higher yield of total FAs in rye bran lipids was obtained compared to that of A-AH. The application of Viscoferm ensured up to a 7.7% higher amount of FAs in wheat bran lipids than those of A-AH. The amount of oleic, linoleic, and linolenic acids in rye bran lipids extracted after EH with Celluclast 1.5 L was 14.0%, 10.7%, and 24.4% higher than in lipids after A-AH. A fairly lower but still relevant difference in the yield of oleic, linoleic, and linolenic acids between EH and A-AH lipids was observed after EH of wheat bran with Viscoferm; up to 10.4%, 10.0%, and 17.0% higher amounts were achieved, respectively. In the context of waste-free technology, the established process of EH can find application within the food industry and can successfully be used for the production of both FAs and fermentable sugars in a more sustainable way.

## Figures and Tables

**Figure 1 biology-11-00721-f001:**
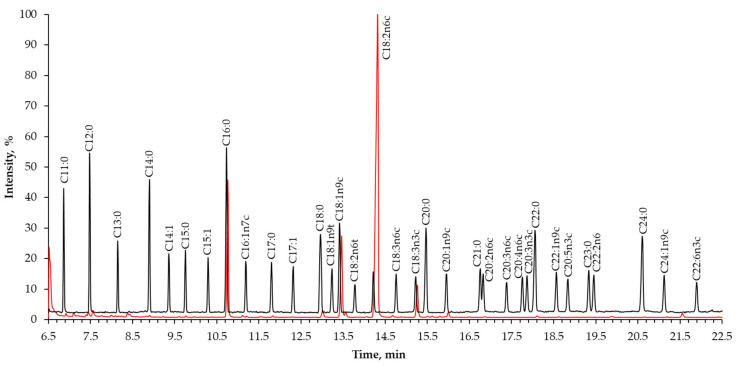
Representative chromatographic separation of C4–C24 fatty acid methyl ester standards (black line) and FAMEs of lipids extracted after enzyme-assisted hydrolysis of rye bran using a Celluclast 1.5 L cellulose-degrading enzyme for 48 h (red line). Samples injection volume 1.0 uL (0.5 ug mL^−1^).

**Figure 2 biology-11-00721-f002:**
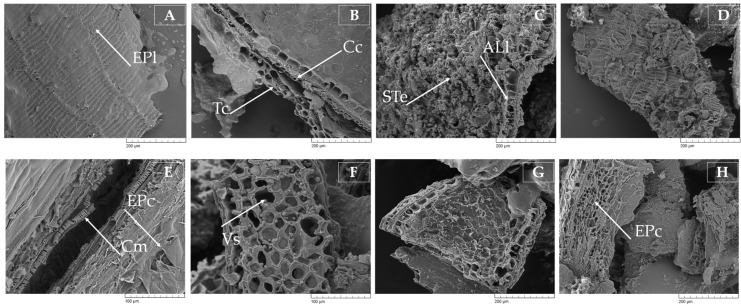
Representative SEM micrographs of untreated wheat (**A**,**B**) and rye (**C**,**D**) and enzymatically hydrolyzed wheat (**E**,**F**) and rye (**G**,**H**) cereal bran samples accomplished by cellulose-degrading commercial multi-enzyme complex Viscozyme L for 48 h. Note: EPl—epidermal layer; Tc—tube cells; Cc—cross cells; STe—starchy endosperm; ALl—aleurone layer; Cm—cellulose microfibers; EPc—epidermal layer cracking; Vs—void spaces in testa and nucellar tissues.

**Figure 3 biology-11-00721-f003:**
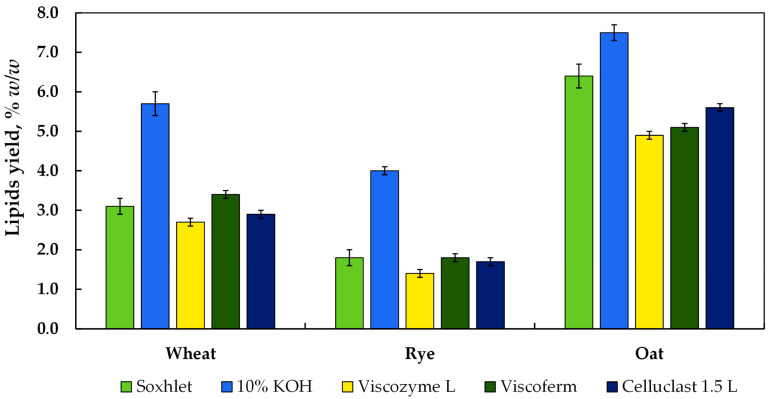
The content of lipids (% *w*/*w*) recovered from wheat, rye, and oat bran samples. Note: Values are means ± SD of triplicates (*n* = 3). SD—standard deviation.

**Table 1 biology-11-00721-t001:** Nutritional compositions of cereal bran by-products derived from rye, wheat, and oat grains, g 100 g^−1^ DW.

	Major Nutrient Profiles, g 100 g^−1^ DW					
Type of Bran	Moisture, %	Crude Carbohydrates	Starch	Crude Lipids	Crude Proteins	Dietary Fiber
Wheat	11.9 ± 0.2 ^a^	20.3 ± 0.4 ^c^	8.7 ± 0.0 ^c^	4.5 ± 0.1 ^b^	16.2 ± 0.4 ^a^	46.5 ± 2.1 ^a^
Rye	11.7 ± 0.2 ^a^	30.9 ± 0.5 ^b^	18.6 ± 0.0 ^b^	3.8 ± 0.1 ^c^	16.9 ± 0.5 ^a^	36.0 ± 1.9 ^b^
Oat	12.4 ± 0.3 ^a^	50.0 ± 0.9 ^a^	47.6 ± 0.9 ^a^	6.7 ± 0.5 ^a^	14.0 ± 0.7 ^b^	14.0 ± 1.7 ^c^

Note: Values are means ± SD values of triplicates (*n* = 3). Means within the same column with different superscript letters (^a, b^, and ^c^) are significantly different at *p* ≤ 0.05; DW—dry weight; SD—standard deviation.

**Table 2 biology-11-00721-t002:** List of commercial cellulose-degrading enzymes applied in this study.

Commercial Enzyme	Declared Activity	Enzyme Activity	Source	EC Number
Viscozyme^®^ L	100 FBG g^−1^	**Endo-1,3-(1,4)-β-d-glucanase**, Endo-1,4-β-xylanase, Non-reducing end α-L-arabinofuranosidase	*Aspergillus aculeatus*	3.2.1.6 3.2.1.8 3.2.1.55
Viscoferm^®^	222 FBG g^−1^	**Endo-1,3-(1,4)-β-d-glucanase**, Endo-1,4-β-xylanase	*Aspergillus* spp.	3.2.1.6 3.2.1.8
Celluclast^®^ 1.5 L	700 EGU g^−1^	**Endo-1,4-β-d-glucanase**	*Trichoderma reesei*	3.2.1.4

Note: The main activities of enzymes used in this study are highlighted in bold. EC—enzyme commission; EGU—endoglucanase units; FBG—fungal β-glucanase units.

**Table 3 biology-11-00721-t003:** The release of mono- and disaccharides after 48 h enzymatic hydrolysis of cereal bran samples using three enzyme complexes, i.e., Viscozyme L, Viscoferm, and Celluclast 1.5 L (g 1000 mL^−1^ of bran hydrolysate).

		Wheat Bran				Rye Bran				Oat Bran		
Carbohydrate	Control	Vzym	Vferm	Cell	Control	Vzym	Vferm	Cell	Control	Vzym	Vferm	Cell
Xyl	0.4 ± 0.0 ^d^	4.6 ± 0.0 ^c^	6.1 ± 0.2 ^b^	8.7 ± 0.1 ^a^	0.2 ± 0.0 ^d^	5.9 ± 0.2 ^c^	7.8 ± 0.3 ^b^	8.4 ± 0.1 ^a^	n.d.	0.8 ± 0.0 ^b^	1.2 ± 0.0 ^b^	2.1 ± 0.0 ^a^
Ara	0.5 ± 0.0 ^b^	2.7 ± 0.0 ^a^	3.2 ± 0.0 ^a^	2.8 ± 0.0 ^a^	0.6 ± 0.0 ^d^	6.2 ± 0.2 ^a^	4.9 ± 0.1 ^b^	1.5 ± 0.0 ^c^	n.d.	0.8 ± 0.0 ^a^	0.9 ± 0.0 ^a^	n.d.
Fru	n.d.	2.1 ± 0.0 ^a^	0.7 ± 0.0 ^b^	0.4 ± 0.0 ^b^	0.2 ± 0.0 ^c^	1.7 ± 0.0 ^a^	0.8 ± 0.0 ^b^	0.5 ± 0.0 ^b^	n.d.	1.6 ± 0.1 ^a^	0.3 ± 0.0 ^b^	n.d.
Glu	n.d.	17.9 ± 0.2 ^a^	7.9 ± 0.1 ^c^	11.3 ± 0.3 ^b^	n.d.	32.4 ± 0.4 ^a^	12.3 ± 0.2 ^c^	15.9 ± 0.3 ^b^	n.d.	47.6 ± 1.3 ^a^	16.1 ± 0.4 ^c^	18.4 ± 0.5 ^b^
Suc	1.9 ± 0.1	n.d.	n.d.	n.d.	2.9 ± 0.1	n.d.	n.d.	n.d.	0.9 ± 0.1	n.d.	n.d.	n.d.
Mal	0.3 ± 0.0 ^c^	0.9 ± 0.0 ^b^	0.9 ± 0.0 ^b^	3.1 ± 0.0 ^a^	0.2 ± 0.0 ^c^	1.9 ± 0.0 ^b^	0.7 ± 0.0 ^c^	2.6 ± 0.0 ^a^	n.d.	1.9 ± 0.0 ^a^	1.1 ± 0.0 ^b^	1.8 ± 0.0 ^ab^
Tot	3.1 ± 0.1 ^c^	28.2 ± 0.2 ^a^	18.8 ± 0.3 ^b^	28.6 ± 0.4 ^a^	4.1 ± 0.1 ^d^	48.6 ± 0.6 ^a^	26.5 ± 0.6 ^c^	28.9 ± 0.4 ^b^	0.9 ± 0.1 ^c^	52.9 ± 1.4 ^a^	19.7 ± 0.4 ^c^	22.9 ± 0.5 ^b^

Note: Values are means ± SD of triplicates (*n* = 3). Xyl—xylose; Ara—arabinose; Fru—fructose; Glu—glucose; Suc—sucrose; Mal—maltose; Tot—total sugar content; Vzym—commercial multi-enzyme preparation Viscozyme L; Vferm—commercial multi-enzyme preparation Viscoferm; Cell—commercial enzyme preparation Celluclast 1.5 L; n.d.—not detected; SD—standard deviation. The concentration of sugars in control CB samples is expressed as g 100 g^−1^ on a dry weight basis. Means within the same carbohydrate and bran type with different superscript letters (^a,b,c,d^) are significantly different at *p* ≤ 0.05.

**Table 4 biology-11-00721-t004:** Fatty acid composition of wheat bran lipids according to the pre-treatment method, mg g^−1^ lipid.

	Wheat									
Fatty Acid	Soxhlet		10% KOH		Vzym		Vferm		Cell	
	Mean	SD	Mean	SD	Mean	SD	Mean	SD	Mean	SD
Myristate C14:0	1.0	0.0	0.8	0.1	1.5	0.0	1.8	0.2	0.9	0.0
Pentadecanoate C15:0	0.8	0.0	1.0	0.1	0.8	0.0	0.8	0.0	0.7	0.0
Palmitate C16:0	127.0	1.9	132.3	3.1	143.9	1.2	142.9	1.9	126.4	2.0
Heptadecanoate C17:0	1.1	0.2	1.4	0.1	1.4	0.0	1.4	0.0	1.1	0.0
Stearate C18:0	8.5	0.3	8.6	0.3	9.6	0.1	9.7	0.0	8.5	0.0
Oleate C18:1 (c9)	104.5	1.1	106.2	2.5	121.3	2.3	117.2	0.9	104.8	1.9
Vaccenate C18:1 (t11)	2.6	0.4	2.3	0.0	2.8	0.2	3.0	0.0	2.0	0.1
Linoleate C18:2 (c9,c12)	365.8	0.0	379.9	7.3	408.6	6.1	417.9	4.2	377.6	10.8
Linolenate C18:3 (c9,c12,c15)	22.0	0.9	21.9	0.4	24.9	0.6	25.7	0.5	23.6	0.8
Arachidate C20:0	2.1	0.0	1.6	0.1	1.2	0.2	1.1	0.2	0.9	0.0
CLA linoleate C18:2 (c9,t11)	0.4	0.0	2.7	0.1	0.8	0.1	1.1	0.0	0.6	0.0
CLA linoleate C18:2 (t10,c12)	0.4	0.1	1.9	0.0	0.5	0.0	0.8	0.0	0.4	0.0
11-eicosenoate C20:1 (c11)	3.3	0.3	3.1	0.1	3.7	0.0	3.7	0.1	3.3	0.1
Eicosadienoate C20:2 (c11,c14)	BLQ	−	0.5	0.1	0.5	0.1	0.6	0.1	0.5	0.0
Behenate C22:0	3.1	0.7	3.3	0.2	2.2	0.2	2.0	0.1	1.7	0.1
Erucate C22:1 (c9,c11)	0.1	0.1	0.1	0.0	0.1	0.0	0.1	0.0	0.1	0.0
Tricosanoate C23:0	1.2	0.2	10.1	1.2	1.0	0.2	0.5	0.0	0.3	0.1
Tetracosanoate C24:0	0.8	0.2	1.1	0.1	0.8	0.1	0.6	0.0	1.3	0.1
**SFA**	145.6	3.6	160.3	5.2	162.5	2.1	160.8	2.5	141.7	2.4
**MUFA**	110.6	1.8	111.7	2.6	127.9	2.5	124.0	1.0	110.3	2.1
**PUFA**	388.6	1.0	406.9	7.9	435.4	6.9	446.1	4.9	402.7	11.7
**∑_MUFA + PUFA_**	499.2	2.9	518.6	10.5	563.3	9.4	570.1	5.9	513.0	13.7
**∑_SFA + MUFA + PUFA_**	644.8	6.5	678.9	15.7	725.8	11.5	730.9	8.4	654.7	16.1
**CLA**	0.8	0.1	4.6	0.1	1.4	0.1	1.9	0.0	1.0	0.0

Note: Values are means ± SD of triplicates (*n* = 3). SFA—saturated fatty acids; MUFA—monounsaturated fatty acids; PUFA—polyunsaturated fatty acids; CLA—conjugated linoleic acid; Vzym—commercial multi-enzyme preparation Viscozyme L; Vferm—commercial multi-enzyme preparation Viscoferm; Cell—commercial enzyme preparation Celluclast 1.5 L; BLQ—below limit of quantification; SD—standard deviation.

**Table 5 biology-11-00721-t005:** Fatty acid composition of rye bran lipids according to the pre-treatment method, mg g^−1^ lipid.

	Rye									
Fatty Acid	Soxhlet		10% KOH		Vzym		Vferm		Cell	
	Mean	SD	Mean	SD	Mean	SD	Mean	SD	Mean	SD
Myristate C14:0	1.3	0.0	0.9	0.0	1.4	0.1	0.9	0.0	1.0	0.1
Pentadecanoate C15:0	0.8	0.0	1.2	0.0	0.8	0.1	0.9	0.1	0.8	0.1
Palmitate C16:0	90.2	1.5	125.5	1.3	117.1	0.6	114.0	2.0	121.5	3.3
Heptadecanoate C17:0	1.2	0.1	1.5	0.1	1.3	0.0	1.4	0.0	1.1	0.0
Stearate C18:0	6.9	0.2	7.2	0.2	6.8	0.3	6.6	0.0	7.5	0.4
Oleate C18:1 (c9)	64.2	1.4	76.5	1.1	71.2	0.1	72.8	0.5	87.2	3.3
Vaccenate C18:1 (t11)	1.7	0.0	2.3	0.1	2.0	0.2	2.0	0.3	2.3	0.2
Linoleate C18:2 (c9,c12)	282.7	6.4	380.6	9.1	341.4	4.6	357.1	5.5	421.4	8.8
Linolenate C18:3 (c9,c12,c15)	21.6	0.5	27.2	0.7	25.0	0.2	26.5	0.6	33.9	0.9
Arachidate C20:0	1.4	0.1	1.7	0.1	0.7	0.0	0.8	0.1	0.8	0.1
CLA linoleate C18:2 (c9,t11)	0.1	0.0	2.1	0.1	0.6	0.0	1.7	0.1	0.9	0.1
CLA linoleate C18:2 (t10,c12)	0.2	0.0	1.6	0.0	0.5	0.0	1.1	0.0	0.5	0.0
11-eicosenoate C20:1 (c11)	3.9	0.3	4.3	0.1	4.0	0.1	4.1	0.2	5.2	0.1
Eicosadienoate C20:2 (c11,c14)	BLQ	−	0.8	0.1	0.5	0.1	0.8	0.0	0.8	0.0
Behenate C22:0	2.7	0.1	2.9	0.1	2.0	0.1	1.8	0.0	1.6	0.1
Erucate C22:1 (c9,c11)	0.4	0.0	0.4	0.0	0.3	0.0	0.4	0.0	0.5	0.0
Tricosanoate C23:0	0.9	0.1	12.9	1.1	0.7	0.0	0.4	0.0	0.3	0.0
Tetracosanoate C24:0	1.1	0.1	1.1	0.1	0.7	0.1	0.6	0.1	4.6	0.0
**SFA**	106.4	2.2	154.9	3.1	131.6	1.2	127.2	2.3	139.2	4.1
**MUFA**	70.3	1.8	83.5	1.4	77.6	0.4	79.3	1.0	95.2	3.6
**PUFA**	304.6	6.9	412.3	9.9	368.1	4.9	387.2	6.2	457.3	9.9
**∑_MUFA + PUFA_**	374.9	8.6	495.8	0.0	445.7	5.4	466.5	7.2	552.5	13.5
**∑_SFA + MUFA + PUFA_**	481.3	10.9	650.7	14.3	577.3	6.5	593.7	9.5	691.7	17.6
**CLA**	0.3	0.0	3.7	0.1	1.2	0.0	2.9	0.1	1.3	0.1

Note: Values are means ± SD of triplicates (*n* = 3). SFA—saturated fatty acids; MUFA—monounsaturated fatty acids; PUFA—polyunsaturated fatty acids; CLA—conjugated linoleic acid; Vzym—commercial multi-enzyme preparation Viscozyme L; Vferm—commercial multi-enzyme preparation Viscoferm; Cell—commercial enzyme preparation Celluclast 1.5 L; BLQ—below limit of quantification; SD—standard deviation.

**Table 6 biology-11-00721-t006:** Fatty acid composition of oat bran lipids according to the pre-treatment method, mg g^−1^ lipid.

	Oat									
Fatty Acid	Soxhlet		10% KOH		Vzym		Vferm		Cell	
	Mean	SD	Mean	SD	Mean	SD	Mean	SD	Mean	SD
Myristate C14:0	2.6	0.0	1.8	0.0	2.3	0.1	2.1	0.1	1.4	0.1
Pentadecanoate C15:0	BLQ	−	0.2	0.0	BLQ	−	BLQ	−	BLQ	−
Palmitate C16:0	150.5	0.1	149.0	0.7	144.4	0.4	137.0	2.4	143.1	0.5
Heptadecanoate C17:0	1.0	0.0	1.0	0.0	0.7	0.0	1.0	0.1	1.0	0.0
Stearate C18:0	16.6	0.2	14.4	0.2	14.0	0.2	12.4	0.2	15.8	0.1
Oleate C18:1 (c9)	343.9	4.8	300.0	1.0	270.6	1.8	266.6	1.3	332.6	0.8
Vaccenate C18:1 (t11)	4.2	0.1	4.7	0.1	3.7	0.0	2.4	0.3	3.2	0.4
Linoleate C18:2 (c9,c12)	323.7	4.5	321.4	2.9	272.9	1.0	272.0	1.3	314.1	3.0
Linolenate C18:3 (c9,c12,c15)	7.7	0.2	7.4	0.2	6.1	0.0	6.2	0.2	7.3	0.0
Arachidate C20:0	0.9	0.8	0.7	0.1	0.5	0.0	0.7	0.1	0.7	0.1
CLA linoleate C18:2 (c9,t11)	0.4	0.0	1.7	0.1	0.3	0.0	0.9	0.0	0.3	0.0
CLA linoleate C18:2 (t10,c12)	0.1	0.0	1.1	0.1	0.3	0.0	0.5	0.0	0.2	0.0
11-eicosenoate C20:1 (c11)	5.7	0.1	4.7	0.0	3.9	0.0	4.4	0.1	5.7	0.0
Eicosadienoate C20:2 (c11,c14)	BLQ	−	BLQ	−	BLQ	−	0.1	0.0	0.1	0.0
Behenate C22:0	1.7	0.2	1.3	1.7	1.6	0.1	1.3	0.1	1.3	0.0
Erucate C22:1 (c9,c11)	0.2	0.0	0.1	0.0	0.1	0.0	0.1	0.0	0.1	0.0
Tricosanoate C23:0	1.0	0.1	1.6	0.7	0.7	0.2	0.7	0.0	0.6	0.0
Tetracosanoate C24:0	0.5	0.0	0.4	0.0	0.4	0.0	0.3	0.0	0.4	0.0
**SFA**	174.8	1.4	170.5	3.7	164.6	1.0	155.5	2.9	164.3	0.9
**MUFA**	353.9	5.1	309.4	1.1	278.4	1.9	273.5	1.7	341.7	1.2
**PUFA**	331.8	4.7	331.7	3.3	279.6	1.1	279.7	1.6	322.0	3.1
**∑_MUFA + PUFA_**	685.7	9.8	641.1	4.4	558.0	3.0	553.2	3.3	663.6	4.3
**∑_SFA + MUFA + PUFA_**	860.6	11.2	811.6	8.1	722.6	4.0	708.7	6.2	828.0	5.2
**CLA**	0.4	0.0	2.8	0.2	0.6	0.0	1.3	0.1	0.4	0.0

Note: Values are means ± SD of triplicates (*n* = 3). SFA—saturated fatty acids; MUFA—monounsaturated fatty acids; PUFA—polyunsaturated fatty acids; CLA—conjugated linoleic acid; Vzym—commercial multi-enzyme preparation Viscozyme L; Vferm—commercial multi-enzyme preparation Viscoferm; Cell—commercial enzyme preparation Celluclast 1.5 L; BLQ—below limit of quantification; SD—standard deviation.

## Data Availability

The data sets and analysis of the study are available from the corresponding author upon reasonable request.

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
