# Peer review of "An Environmentally Friendly Approach for the Release of Essential Fatty Acids from Cereal By-Products Using Cellulose-Degrading Enzymes"

_biology, 2022, doi:10.3390/biology11050721_

Round 1

Reviewer 1 Report

Summary

According to the authors, the main intention of the present work was to investigate the ability of cellulose-degrading enzymes to release fatty acids from complex matrices of cereal by-products during enzymatic hydrolysis.

The results revealed that, in the context of waste-free technology, the established process of enzymatic hydrolysis can find application within the food industry and successfully be used for the production of both fatty acids and fermentable sugars in a more sustainable way.

Major Issues

The main problem is that I do not fully understand the enzymatic hydrolysis experiment (section 2.6.). I do not understand what is meant when only endo-1,4-β-xylanase activity is mentioned in the cases of Viscozyme L and Viscoferm (lines 185-186). And I do not understand why only one pH (4.6) and one temperature (44 ± 1 °C) are mentioned, knowing that we have different enzymes (or commercial preparations) and that each of them will have different activity and stability optima, with the extraordinary influence that this entails on the results. I assume that the experiments have been done at different pHs and temperatures, and that the authors have forgotten to mention them.

Minor Issues

They are attached in the manuscript itself.

Author Response

Response to Reviewer’s 1 comments

R: According to the authors, the main intention of the present work was to investigate the ability of cellulose-degrading enzymes to release fatty acids from complex matrices of cereal by-products during enzymatic hydrolysis.

The results revealed that, in the context of waste-free technology, the established process of enzymatic hydrolysis can find application within the food industry and successfully be used for the production of both fatty acids and fermentable sugars in a more sustainable way.

A: The authors would like to thank you for careful and thorough checking of our manuscript and valuable comments. In preparing the manuscript authors have incorporated most of the changes suggested. The authors refer to them in detail below.

Major issues: 

R: The main problem is that I do not fully understand the enzymatic hydrolysis experiment (section 2.6.). I do not understand what is meant when only endo-1,4-β-xylanase activity is mentioned in the cases of Viscozyme L and Viscoferm (lines 185-186). And I do not understand why only one pH (4.6) and one temperature (44 ± 1 °C) are mentioned, knowing that we have different enzymes (or commercial preparations) and that each of them will have different activity and stability optima, with the extraordinary influence that this entails on the results. I assume that the experiments have been done at different pH and temperatures, and that the authors have forgotten to mention them.

A: The authors understand the reviewer's concern about the methodology described in section: 2.6., and the authors wish to clarify the situation regarding this misunderstanding. In choosing pH and temperature the authors were guided by preliminary research made in past working with cellulose-degrading enzymes considering the pH and temperature on the yield of cellulose and hemicellulose degradation products (individual sugars specifically). The authors dare to assure the reviewer that we observed no significant differences in the content of these products in the obtained hydrolysates working under the pH range of 3.5-5.5. So this was the main reason the authors to stay with pH 4.6 which is indeed close to optimal for most the glucanases.

Besides, the selection of EH reaction time has been based on our previous study aimed at releasing ferulic acid, the highest yield of which was achieved after 48h performing EH with Viscozyme L, Vidcoferm, and Celluclast 1.5 L. 

Minor issues:

R: They are attached in the manuscript itself.

A: The authors have gone through the entire manuscript and amended shortfalls and typos according to reviewer's suggestions. Please refer to the manuscript. 

R: P3, lines 112-114: I can´t understand this sentence.

A: Dear reviewer! The authors apologies for being careless in expression their thoughts. This fragment has been corrected and is now presented in the manuscript as: According to morphological assessment, such hydrological layers as inner pericarp (tube cells, cross cells), outer pericarp, aleurone, and hyaline layers with attached starch granules, seed coat (testa) were distinguished in the bran samples.

R: P4, line 143: I assume it was used for the derivatization of the standard solution, but it is not mentioned anywhere.

A: The authors’ apologies for misleading the reviewer regarding these reagents. The authors clarify that in this work we used TMPAH reagent in the presence of methanol was used as a derivatization agent, and pyridine and BF3 have been included in the list erroneously. We have removed these reagents.

R: P4, lines 151-152: Not in the case of Celluclast, one single enzyme has been declared.

A: The authors agree with the reviewer observation. Now this information is corrected in the manuscript.

R: P4, lines 155-156: I would change the order to the order of the third column: 3.2.1.4 / 3.2.1.8 / 3.2.1.55.

A: The Table 2 has been fully revised. Please refer to the corrected version of the manuscript.

R: P4-P5, lines 180-181: So why was the same pH (4.6) and temperature (44 ± 1 °C) used in all cases?

A: Dear reviewer! Please refer to our reply regarding the major issue on page 1. We hope that provided explanation resolves this misunderstanding.

R: P5, line 186: In Table 2 the name "endo-1,3-(1,4)-β-D-glucanase" is used. One or the other, but use always the same name.

A: The authors apologize for being careless and assure that the main activity of Celluclast 1.5 L is endo-1,4-β-D-glucanase, which now is used within the entire manuscript.

R: P5, line 196: Is this name correct?

A: Yes, this name is correct, since it is a literal translation from German “Osterode am Harz”. To not mislead the readers, the authors decided to use the more common name of this city “Osterode”.

R: P6, line 241-242: I think there is some mistake here.

A: The authors appreciate valuable observation by the reviewer very much. This fragment has been corrected.

R: P8, line 334: In starch, the glycosidic bonds have an alpha configuration, so beta-glucanase activity has no effect.

A: Dear reviewer! This fragment has been corrected and is now presented in the manuscript as: A considerably higher amount of glucose in oat bran hydrolysates is due to the presence of β-glucans, which alongside cellulose ensures the release of glucose monomers by endo-1,3-(1,4)-β-D-glucanase.

R: P8, lines 338-339: But that's mainly because more units of activity were added, and the pH and temperature of the experiment were closer to the optima for Viscozyme L activity and stability ( ≈ Aspergillus aculeatus), right?

A: The authors agree with the reviewer's observation that the amount of enzyme (Celluclast 1.5 L) added to the CB substrate is higher, however, the difference is only 4 units, i.e. 6 FBG for Viscozyme L and Viscoferm, and 10 EGU for Celluclast 1.5 L. The authors believe that this is not a reason for such a big difference in the number of released sugars.  

R: P11. Lines 387-388: There must be some mistake in this sentence.

A: The authors have revised this fragment.

R: P12, lines 421-426: But what are the optimum pHs and temperatures of activity and stability for each of the 3 commercial preparations? If those of the experiment are more in line with those of Celluclast, why shouldn't that be the main reason?

A: Dear reviewer! The authors assure that no significant effect has been observed under the pH of 3.5-5.5 in our preliminary trials with EH of CB. The authors decided to stay with pH 4.6 which is close to optimal for most the glucanases.

We also revised this fragment and now it appears as follows: As seen, multi-enzyme complex Viscozyme L in the case of wheat bran samples demonstrated better hydrolytic performance since up to 16.1% and 14.3% higher yield of oleic acid was recovered compared with the Soxhlet extraction and A-AH hydrolysis, respectively. However, considering the yield of oleic acid using other enzymes no statistically significant difference (p ≥ 0.05) in the content of this FA was observed. However, considering the amount of oleic acid in wheat lipids using other enzymes no statistically significant difference (p ≥ 0.05) in the content of this FA was observed. The superior efficiency of hydrolysis and release of oleic acid was noticed after EH of rye bran with Celluclast 1.5 L. As seen, up to 35.7% and 14.0% higher yield was reached compared with Soxhlet extraction and A-AH hydrolysis, respectively. The amount of oleic acid released by the activity of Celluclast 1.5 L was found to be significantly (p ≤ 0.05) higher than that obtained after A-AH or EH with Viscozyme L and Visconfirm. It is worth noting, though, that the release of oleic acid was not correlated with the amount of sugars in hydrolysates, presumably revealing that this FA in rye bran matrix presented as a glycolipid complex where FA attached to carbohydrates by a 1,4-β-D-glycosidic bond. The release of oleic acid from rye bran matrix most likely is associated with glycoside hydrolase activity (endo-1,4-β-D-glucanase) that was rather intense in Celluclast 1.5 L than in Viscozyme L and Viscoferm.

R: P12, lines 427-430: So why is it that in the case of wheat the lowest yield is obtained using Celluclast?

A: Dear reviewer. The authors have reconsidered this observation and now it appears as follows: As seen, multi-enzyme complex Viscozyme L in the case of wheat bran samples demonstrated better hydrolytic performance since up to 16.1% and 14.3% higher yield of oleic acid was recovered compared with the Soxhlet extraction and A-AH hydrolysis, respectively. However, statistical analysis did not reveal significant differences (p ≥ 0.05) in the content of this FA between hydrolytic enzymes used.

R: P14, lines 470-471: In starch, the glycosidic bonds have an alpha configuration, so beta-glucanase activity has no effect.

A: Dear reviewer! This fragment has been amended and is now presented in the manuscript as follows: In general, higher glycoside hydrolase activity (endo-1,4-β-D-glucanase) declared for Celluclast 1.5 L led to a substantial breakdown of glycosidic bonds between N-SPs and lipids, which presumably promoted the release of FAs.

R: P14, line 472: But why not in wheat?

A: Dear reviewer! We already answered this question earlier. Please refer to “R: P12, lines 427-430:”.   

R: P14, 479-480: But where does it come from that oat bran lipids have fewer FAs? I see just the opposite. What does the fact that extractability is not improved by hydrolysis have to do with the material having fewer FAs? Either I am misunderstanding, or you are mixing up different things.

A: Dear reviewer! Certainly, oat bran lipids contained the highest amount of FAs. The authors intended to say that the yield of FAs after EH of CB was significantly lower (from 772.6 to 828.0 mg) than that obtained after Soxhlet extraction (860.6 mg). The authors apologize for the inaccuracy in the expressions.

By “extractability” the authors meant the release and yield FAs from the CB matrix.

R: P17, lines 506-509: This is repeated.

A: The authors agree with the reviewer's observation. This fragment has been omitted.

R: P17, line 510: In starch, the glycosidic bonds have an alpha configuration, so beta-glucanase activity has no effect.

A: Dear reviewer! The authors have revised this fragment and now it appears as follows: The results showed that the action of C-DE promoted the release of subcellular organelles and the breakdown of glycosidic bonds between N-SPs and lipids in glycolipids, thus contributing to better transfer of FAs into the extraction solvent.

A: The authors hope that the above explanations and adherence to the suggestions made in the reviews will render the attached manuscript appropriate and free from any understatement.

On behalf of all the co-authors

Yours sincerely,

Vitalijs Radenkovs

Principal investigator, Institute of Horticulture (LatHort)

Reviewer 2 Report

Thank you for granting me the opportunity to review this piece of work. In this work, Radenkovs et al. investigated the ability of cellulose-degrading enzymes to release fatty acid from cereal by-products including wheat, rye, and oat brans using the enzymes Viscozyme L, Viscoferm, and Celluclast 1.5 L. Kindly, find below my comments for your response.

Abstract

Line 19: The authors should expand the abbreviation “FFA”.

Line 20: Replace “confronted” with “compared”.

Introduction

Line 53: kindly replace “physical-chemical” with “physico-chemical”

Line 66: Remove “back”

Line 68: Revise this “There have hitherto developed various pretreatment………….”

Line 68-71: Revise the sentence

Line 84: replace “proved” with “proven”

Materials and Methods

Line 110: replace “gathered” with “obtained”

Line 111: Shouldn’t “…..Ltd. Voldemars” rather be “….Voldemars Ltd.”

Line 150: replace “have been” with “was”

Line 159: revise “…about of…” to “…..of about….”

Line 190: what type of thermal processing was used?

Line 262: add degree Celsius to “250”

Results and Discussion

Line 377-378: Please, revise the sentence

Line 383: Please, revise the sentence

Table 5. The authors have indicated a foot note “Note: Please see Table 4”. However, Tables must always stand alone. Therefore, the authors should expand the abbreviations

Also, in Table 4, the authors indicate the “Standard Deviation” with the abbreviation “SD”. However, in Fig. 5 and 6, the authors use “STDDEV”. I suggest the authors use “SD” and expand that under each Table.

Conclusion

Kindly, reduce the number of words for the conclusion. Let it address specifically the objectives of the work.

Author Response

Response to Reviewer’s 2 comments

R: Thank you for granting me the opportunity to review this piece of work. In this work, Radenkovs et al. investigated the ability of cellulose-degrading enzymes to release fatty acid from cereal by-products including wheat, rye, and oat brans using the enzymes Viscozyme L, Viscoferm, and Celluclast 1.5 L. Kindly, find below my comments for your response.

A: The authors would like to thank you for careful and thorough checking of our manuscript and valuable comments. In preparing the manuscript authors have incorporated most of the changes suggested. The authors refer to them in detail below.

R: Line 19: The authors should expand the abbreviation “FFA”.

A: The authors apologize for misleading the reviewer with such an abbreviation. The authors meant FAs which refer to fatty acids.

R: Line 20: Replace “confronted” with “compared”.

A: The word “confronted” has been substituted with “compared”.

Introduction

R: Line 53: kindly replace “physical-chemical” with “physico-chemical”

A: The word “physical-chemical” has been substituted with “physico-chemical”.

R: Line 66: Remove “back”

A: The word “back” was removed.

R: Line 68: Revise this “There have hitherto developed various pretreatment………….”

A: Many thanks for your valuable remark. The authors have revised this fragment and it appears as follows: “There have been various pretreatment strategies developed to date aimed at valorization of CB to food and pharmaceutical applications…”.

R: Line 68-71: Revise the sentence

A: This sentence has been revised.

R: Line 84: replace “proved” with “proven”

A: The word “proved” has been substituted with “proven”.

Materials and Methods

R: Line 110: replace “gathered” with “obtained”

A: The word “gathered” has been substituted with “obtained”.

R: Line 111: Shouldn’t “…..Ltd. Voldemars” rather be “….Voldemars Ltd.”

A: Corrected.

R: Line 150: replace “have been” with “was”

A: The tense “have been” has been substituted with “was”.

R: Line 159: revise “…about of…” to “…..of about….”

A: Revised.

R: Line 190: what type of thermal processing was used?

A: By thermal processing, the authors meant heating hydrolysates for 10 min under 100 °C in a water bath.

R: Line 262: add degree Celsius to “250”

A: Dear reviewer! The authors decided to remove plus sign from temperatures that are above 0 °C. It is obvious that 250 °C means +250 °C. We hope the reviewer understood our intention.

Results and Discussion

R: Line 377-378: Please, revise the sentence

A: Many thanks for your valuable observation. The results describing lipids yield have been revised.

R: Line 383: Please, revise the sentence

A: This section has been fully revised.

R: Table 5. The authors have indicated a foot note “Note: Please see Table 4”. However, Tables must always stand alone. Therefore, the authors should expand the abbreviations

A: Dear reviewer! The authors appreciate your valuable observation. Footnotes now are given under each table.

R: Also, in Table 4, the authors indicate the “Standard Deviation” with the abbreviation “SD”. However, in Fig. 5 and 6, the authors use “STDDEV”. I suggest the authors use “SD” and expand that under each Table.

A: Many thanks for your remark. Now in all tables, the abbreviation for standard deviation SD is given by SD.

Conclusion

R: Kindly, reduce the number of words for the conclusion. Let it address specifically the objectives of the work.

A: The number of words in Conclusion section has been reduced.

The authors hope that the above explanations and adherence to the suggestions made in the reviews will render the attached manuscript appropriate and free from any understatement.

On behalf of all the co-authors

Yours sincerely,

Vitalijs Radenkovs

Principal investigator, Institute of Horticulture (LatHort)

Round 2

Reviewer 1 Report

I believe the manuscript has been improved. However, some mistakes remain:

  • New line 212: "MeOH" is repeated, or something is missing...
  • New line 246: 1.0 min is not a flow rate. The unit is wrong.
  • New line 310: "cellulose" is wrong.
  • New lines 423-424: "A higher lipid yield compared to other enzymes was obtained after EH of oats with Viscoferm,...". It is not correct, it is higher with Celluclast.
  • New line 467: "Visconfirm" is not correct.
  • Some comments made in the old version (lines 437 and 438 of the old version, Tables 4, 5 and 6) have not been taken into account.
  • The comment made on line 479 of the old version has been answered, BUT NO CHANGES HAVE BEEN MADE TO THE TEXT OF THE ARTICLE.

Author Response

R: I believe the manuscript has been improved. However, some mistakes remain:

A: Dear reviewer! The authors appreciate your valuable comments and contribution to this work very much. Certainly, it helped us considerably improve the quality of our work and overcome shortcomings and typing errors.

R: New line 212: "MeOH" is repeated, or something is missing...

A: The repeated word has been removed.

R: New line 246: 1.0 min is not a flow rate. The unit is wrong.

A: The authors apologize for being inaccurate with units. The unit for the flow rate has been changed to 1.0 mL min−1.

R: New line 310: "cellulose" is wrong.

A: The word “cellulose” was substituted with “cellulase”

R: New lines 423-424: "A higher lipid yield compared to other enzymes was obtained after EH of oats with Viscoferm,...". It is not correct, it is higher with Celluclast.

A: The authors have corrected this error.

R: New line 467: "Visconfirm" is not correct.

A: The authors have corrected this error.

R: Some comments made in the old version (lines 437 and 438 of the old version, Tables 4, 5 and 6) have not been taken into account.

A: From the previous review report: “R: Also, in Table 4, the authors indicate the “Standard Deviation” with the abbreviation “SD”. However, in Fig. 5 and 6, the authors use “STDDEV”. I suggest the authors use “SD” and expand that under each Table.”

A1: Now the explanation for the “SD” abbreviation has been provided and expanded under each Table.

R: The comment made on line 479 of the old version has been answered, BUT NO CHANGES HAVE BEEN MADE TO THE TEXT OF THE ARTICLE.

A: If the authors understood you correctly it goes that your comment in the previous version was: “R: Kindly, reduce the number of words for the conclusion. Let it address specifically the objectives of the work.”

A1: Dear reviewer! We hope that now the number of words in this section is sufficient to stress exceptionally the objectives of the present work.

On behalf of all the co-authors

Yours sincerely,

Vitalijs Radenkovs

Principal investigator, Institute of Horticulture (LatHort)